# ISOTROPIC CONTEXTUAL REPRESENTATIONS THROUGH VARIATIONAL REGULARIZATION

## ABSTRACT

Representations from common pretrained language models have been shown to suffer from the degeneration problem, i.e. they occupy a narrow cone in latent space. This problem can be addressed by enforcing isotropy in latent space. In analogy to variational autoencoders, we suggest applying a token-level variational loss to a Transformer architecture and optimize the standard deviation of the prior distribution in the loss function as model parameter to increase isotropy. The resulting latent space is complete and interpretable: any given point is a valid embedding and can be decoded into text again. This allows for text manipulations directly in latent space. Surprisingly, features extracted at sentence-level show competitive results on benchmark classification tasks.

## 1 INTRODUCTION

Self-supervised, attention-based language models are prone to overfitting or memorizing input data, especially when trained on smaller datasets. Nevertheless, models such as BERT (Devlin et al., 2019), based on an encoder-only architecture and trained on very large datasets, exhibit state-of-the-art performance on common natural language processing tasks. The resulting token representations of these models are distributed and highly contextual, but the latent space does not exhibit additional structural properties such as isotropy and resulting completeness (Ethayarajh, 2019).

We develop an autoencoder architecture that avoids overfitting by increasing latent space isotropy. At the same time, enforcing a complete latent space leads to a decoder that is able to interpret (i.e. decode) any given point in latent space. While the primary goal are tasks that require paraphrase generation, the resulting encoders surprisingly also perform on par in transfer tasks with smaller training setups.

Previous work has shown that isotropic latent representation, i.e. distributed across latent space instead of populating a narrow cone (Ethayarajh, 2019), can improve the performance of language models on transfer tasks. Gao et al. (2019) improve the performance of a Transformer-based model with increased isotropy by directly optimizing the cosine distance between the latent representations. Wang et al. (2020) suggest to control the singular value distribution of the output representation matrix and Li et al. (2020a) transform the original representation distribution into a Gaussian distribution through normalizing flows to increase the isotropy of the underlying models.

Similar to Gao et al. (2019), we propose to apply a regularizing constraint during training to enforce an isotropic and, even more, complete latent space. A common regularization network is the so-called Variational Autoencoder (VAE) (Kingma & Welling, 2014). VAE networks consist of an encoder that maps given input data not to a point in latent space but a distribution. A VAE's decoder is required to successfully reconstruct the original input from samples of the latent distribution. Thus, the VAE is optimized to reconstruct the given input sequence whilst enforcing the latent distributions to match a given prior distribution.

We propose a Variational Auto-Transformer (VAT), a VAE based on a Transformer architecture, where the variational loss is computed at token level. We show that by enforcing a Gaussian distribution as latent prior, the latent token-level representations become more isotropic in comparison to models such as BERT. We introduce the prior distribution's standard deviation as model parameter to optimize isotropy and balance the language generation variety against the network's reconstruction ability. Besides isotropy, the completeness of the latent space is demonstrated. The pretrained

decoder can map back every point in latent space to a textual representation, be it actual encoded or synthetic, sampled latent points. This allows for text generation through "variational sampling" and other manipulations such as interpolation directly in latent space. While the token-level representations can be used to generate or manipulate text, we also show that sentence-level representations, e.g. obtained through averaging, are suitable for sentence classification tasks.

## 2 BACKGROUND

An autoencoder is a neural network architecture consisting of an encoder $enc$, that maps any input $\boldsymbol{x} \in \mathbb{R}^d$ to a point $\boldsymbol{z} = enc(\boldsymbol{x})$ in latent space, and a decoder $dec$, such that $\boldsymbol{x} \approx dec(enc(\boldsymbol{x}))$ (Goodfellow et al., 2016). The chosen network architecture defines the families $E$ and $D$ of the encoder and decoder. During training of the autoencoder, the network parameters are optimized to find the optimal $(enc^*, dec^*)$ pair that minimizes a given loss function $\mathcal{L}$:

$$(enc^*, dec^*) = \underset{(enc, dec) \in E \times D}{\arg\min} (\mathcal{L}) \tag{1}$$

For the standard autoencoder with continuous inputs $\boldsymbol{x} \in \mathbb{R}^d$, the reconstruction loss $\mathcal{L} = \mathcal{L}_{REC}$ is:

$$\mathcal{L}_{REC} = \|\boldsymbol{x} - dec(enc(\boldsymbol{x}))\|_2 \tag{2}$$

Over-complete autoencoder architectures with many degrees of freedom are prone to a particular form of overfitting: The encoder maps each data point $\boldsymbol{x}$ to an isolated point $\boldsymbol{z}$ in the latent space such that the decoder memorizes the lossless reconstruction of each of these codes. This highly discrete latent space lacks completeness and is of limited use for advanced NLP applications.

### 2.1 VARIATIONAL AUTOENCODER

The variational autoencoder (VAE) (Kingma & Welling, 2014) can be thought of as generative autoencoder. A VAE encodes an input data point as a distribution over the latent space by adding a regularization term to the loss function. The regularization term $\mathcal{L}_{REG}$ assesses the Kulback-Leibler divergence between the latent distribution and a standard Gaussian based on their means $\boldsymbol{\mu}$ and covariances $\Sigma$:

$$\mathcal{L}_{REG} = -D_{KL}[N(\boldsymbol{\mu}, \Sigma), N(\boldsymbol{0}, I)] \tag{3}$$

$\boldsymbol{\mu}$ and $\Sigma$ are either estimated using standard point estimators from the encoder outputs, or computed by learned functions such that $\boldsymbol{\mu} = g(\boldsymbol{x})$ and $\Sigma = h(\boldsymbol{x})$ with $g \in G$ and $h \in H$, where $G$ and $H$ are families of network architectures. $G$ and $H$ can contribute to the desired properties of the latent representation by implicitly implementing dimensionality reduction or feature disentangling.

Decoding from the latent representation requires sampling from $\boldsymbol{z} \sim N(\boldsymbol{\mu}, \Sigma)$. In order to enable backpropagation despite this sampling operation, Kingma and Welling (Kingma & Welling, 2014) introduced the reparameterization trick: Instead of sampling from the latent distribution, a random sample $\boldsymbol{\epsilon} \sim N(\boldsymbol{0}, I)$ from a standard Gaussian is drawn and then transformed by the computed mean and standard deviation $\boldsymbol{z} = L^T \boldsymbol{\epsilon} + \boldsymbol{\mu}$, where $L^T$ is the Cholesky factor of $\Sigma$.

The VAE's total loss is the sum of the regularization and reconstruction term:

$$(enc^*, dec^*) = \underset{(enc, dec) \in E \times D}{\arg\min} (\mathcal{L}_{REC} + \mathcal{L}_{REG})$$

$$= \underset{(enc, dec) \in E \times D}{\arg\min} \left( \|\boldsymbol{x} - dec(\boldsymbol{z})\|_2 - D_{KL}[N(\boldsymbol{\mu}, \Sigma), N(\boldsymbol{0}, I)] \right) \tag{4}$$

## 3 VARIATIONAL AUTO-TRANSFORMER

Initially proposed for machine translation, the Transformer (Vaswani et al., 2017) can be used as language model and framed as autoencoder. Extended with a token-level variational loss, the VAT

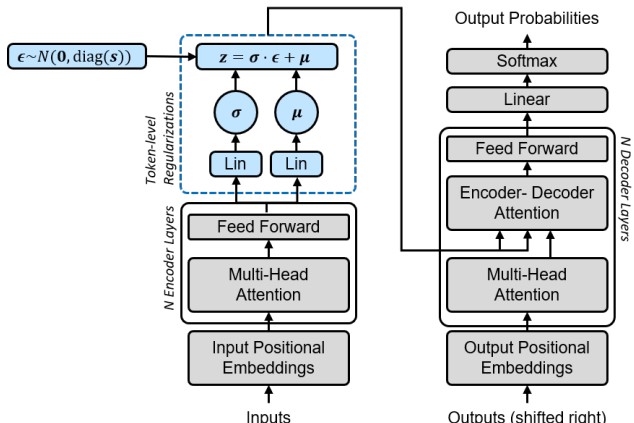

Figure 1: The proposed Variational Auto-Transformer architecture: Mean and covariance for the variational loss (blue, dashed lines) are computed through two independent linear layers. The encoder and decoder (grey, solid lines) architecture is the same as in the original Transformer.

maps input tokens to context-aware, distributed latent representations. Figure 1 illustrates the architecture of the VAT. The set of encoders $E$ and decoders $D$ is defined by the Transformer's network architecture based on self-attention modules. The encoder maps the embedded input sequence $X = \{\boldsymbol{x}_1, \boldsymbol{x}_2, \ldots, \boldsymbol{x}_T\}$, $X \in \mathbb{R}^{T \times d}$ to a latent representation $Z = \{\boldsymbol{z}_1, \boldsymbol{z}_2, \ldots, \boldsymbol{z}_T\}$, $Z \in \mathbb{R}^{T \times d}$, where $T$ is the sequence length and $d$ is the model dimension. To do so, the model predicts $\boldsymbol{mu}_t$ and $\boldsymbol{\sigma}_t$ from $\boldsymbol{x}_t$ and samples $\boldsymbol{z}_t \sim N(\boldsymbol{\mu}_t, \boldsymbol{\sigma}_t)$. $Z$ is then used as attention source in the decoder to predict the next word based on the already produced output. As $X$ and $Z$ depend on $T$, the objective is to minimize

$$\mathcal{L} = \frac{1}{T} \sum_{t=1}^{T} \left( \mathbb{E}_{q(Z|X)} \|\boldsymbol{x}_t - dec(\boldsymbol{z_t})\|_2 - D_{KL}\left[q(\boldsymbol{z_t}|X), p(\boldsymbol{z_t})\right] \right) \tag{5}$$

where $p(\boldsymbol{z}_t) \sim N(\boldsymbol{0}, I)$ and $q(\boldsymbol{z}_t|X) \sim N(\boldsymbol{\mu}_t, \Sigma_t)$. More precisely, we let $\Sigma_t = \text{diag}(\exp \boldsymbol{\sigma}_t)$ where the vector $\boldsymbol{\sigma}_t = (\sigma_j)_{j=1\ldots d}$ is predicted, so that computing the actual value of $\Sigma$ involves the exponential as a non-linear activation function.

The output of the encoder is passed to two linear layers in parallel. The linear layers have the same number of input and output nodes and learn a transformation to predict $\boldsymbol{\mu}_t$ and $\boldsymbol{\sigma}_t$ for each token. The latent representation $Z$ is used as input in the encoder-decoder attention layer of the Transformer's decoder.

## 3.1 SCALING THE REGULARIZATION LOSS

During our experiments, we experienced overfitting on the training data for the reconstruction loss. The learning curves of a corresponding experiment are illustrated in Figure 2a. Weighting the regularization loss according to a logistic annealing function as suggested by Bowman et al. (2016) or by a scaling factor $\beta$ similar to the scaling of the beta-VAE (Higgins et al., 2017), but with $\beta < 1$, only had little effect.

Thus, we propose to scale the covariance matrix of the target distribution of $\mathcal{L}_{REG}$ instead. This is motivated by the observation that standard Gaussians in $d$-dimensional latent spaces (e.g. $d \geq 128$) will overlap considerably if their mean values are - at the same time - regularized to zero. As a consequence, $\mathcal{L}_{REG}$ cannot be minimized by the model without increasing $\mathcal{L}_{REC}$ in an undesirable way. Scaling the standard deviation to a too small value, though, might result in peaky distributions, having no regulatory effect and resulting in a trivial reconstruction objective.

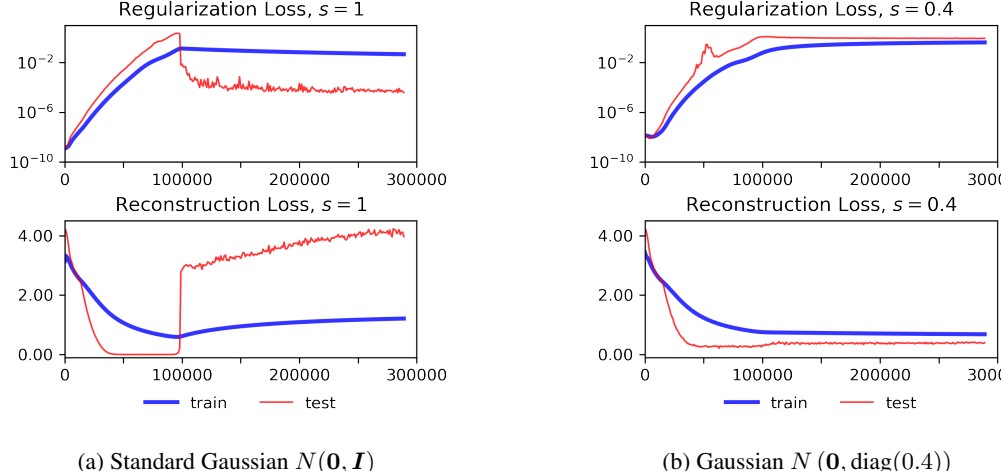

(a) Standard Gaussian $N(\mathbf{0}, \boldsymbol{I})$          (b) Gaussian $N(\mathbf{0}, \mathrm{diag}(0.4))$

Figure 2: Weighted regularization (top) and reconstruction loss (bottom) for train and test data over several epochs for two different Gaussians applied as prior distribution for the regularization term.

We adapt the computation of the regularization loss to incorporate a scaling factor $s$ for the standard deviation as hyperparameter. The closed form of the VAE loss function[1] with a prior standard deviation $\sigma_p = s \in (0, 1]$ and prior $\mu_p = 0$ for a specific token's layer outputs $\boldsymbol{\sigma}_t$ and $\boldsymbol{\mu}_t$ becomes

$$- D_{KL}\left[N(\boldsymbol{\mu}_t, \boldsymbol{\sigma}_t), N(\mathbf{0}, diag(s))\right] = \sum_{i=1}^{d} \frac{1}{2} \log(\sigma_{t,i}^2) - \frac{\sigma_{t,i}^2 + \mu_{t,i}^2}{2s^2} - \log(s) + \frac{1}{2} \quad (6)$$

This results in codes $\boldsymbol{z}_t = \boldsymbol{\sigma}_t \cdot \boldsymbol{\epsilon}_t' + \boldsymbol{\mu}$ being distributed according to a Gaussian with zero mean and a predefined standard deviation, $\boldsymbol{\epsilon}' \sim N(\mathbf{0}, \mathrm{diag}(\boldsymbol{s}))$. The optimal balance between $\mathcal{L}_{REC}$ and $\mathcal{L}_{REG}$ as expressed by the value of $s \in (0, 1]$ is related to the representations' isotropy and will be determined experimentally.

## 4 EXPERIMENTS

The aim of the experiments is to balance the two loss terms. $\mathcal{L}_{REC}$ should be low to maintain the reconstruction ability of the network, while a too small scaling value $s$ implies that the VAT essentially behaves as regular AE. At the same time, we want to optimize the isotropy of the latent representations. In order to choose the best setting, we observe both, the resulting loss values and the properties of the latent space in terms of similarity between representations.

Our VAT model architecture is smaller than the original Transformer architecture, as it is not intended for machine translation. The VAT consists of $N = 4$ encoder and decoder layers, respectively, with $H = 8$ attention heads each and a model dimensionality of $d = 128$. A logistic annealing function (Bowman et al., 2016) is applied to weight $\mathcal{L}_{REG}$. The VAT is trained on the train split of the WMT19 de-en dataset (Barrault et al., 2019) using English sentences only. WMT19 contains data from the news commentary, Wiki titles, Europarl, ParaCrawl and Common Crawl corpora. The data is tokenized using subword tokenization with a target vocabulary of $2^{15}$ tokens. The full details on model and training parameters are listed in the appendix.

### 4.1 OPTIMAL SCALING PARAMETER

The isotropy of the learned representations $\boldsymbol{z}$ is assessed as a function of the scaling factor $s \in (0, 1]$. Ethayarajh (2019) introduced the notion of isotropy of the latent space as the mean cosine similarity between vector representations of random tokens. In an isotropic latent space, representations of randomly sampled tokens have low cosine similarity and do not cluster in a specific direction.

---

[1]Refer to Odaibo (2019) for the derivation of the closed form.

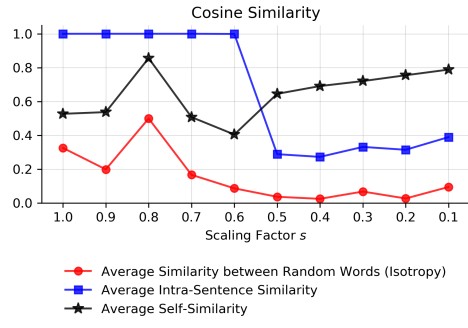 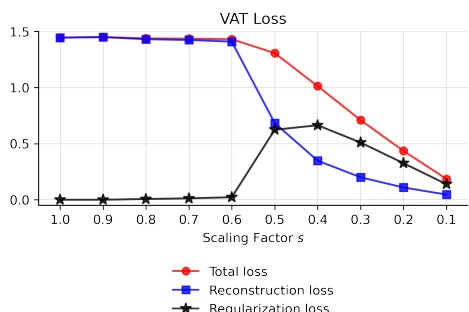

(a) Cosine similarities of the latent representations of the VAT at the end of the training process denoting the isotropy of the latent space and the contextuality of token representations.

(b) Reconstruction loss, weighted regularization loss and total loss of the VAT at the end of the training process.

Figure 3: Optimizing the scaling factor $s$ with respect to isotropy and loss values.

In addition to isotropy, Ethayarajh (2019) introduces the notion of self-similarity and intra-sentence similarity. A high self-similarity, i.e. mean cosine similarity between representations of the same token in different sentences, indicates that neighboring representations capture similar concepts which contributes to the smoothness of the latent space. Intra-sentence similarity measures the mean cosine similarity of tokens occurring in the same sentence to the mean sentence representation, thus being related to contextuality.

BERT exhibits low isotropy (mean cosine similarity $> 0.4$) in its last layers (Ethayarajh, 2019), a phenomenon known as representation degeneration (Gao et al., 2019). In contrast to BERT with high contextuality with respect to the low isotropy, we expect a more complete latent space for the VAT and thus high isotropy. The results in Figure 3a[2] are computed for the STS12 datasets Agirre et al. (2012). The average cosine similarity between randomly sampled words is low for smaller $s$ values, suggesting an isotropic latent space with representations distributed across space. The latent space is most isotropic at $s = 0.4$.

The average self-similarity of tokens greater than $0.6$ suggests a mapping of similar concepts into the same region, indicating smoothness in the latent space. The intra-sentence similarity for $0.5 \geq s \geq 0.1$ is lower than the self-similarity, but still above the similarity of random words. This indicates that contextual information is captured in the representations. Interestingly, for $1 \geq s \geq 0.6$, the token representations seem to "collapse" into an identical representation for all positions in a sequence resulting in an intra-sentence similarity of $1$. Among the words with most context-specific representations are mainly stopwords as they have lowest self-similarity.

In Figure 3b, we report the final loss values after training the models for 3 epochs with different scaling factors $s \in (0, 1]$. Scaling values $1 \geq s \geq 0.6$ result in near zero regularization losses and high reconstruction error. The original input cannot be reproduces any more. This observation corresponds to the findings from Figure 3a, where the representations within a sentence are identical. Starting from $s = 0.5$, the reconstruction loss drops and with $s = 0.4$, the reconstruction loss falls below the regularization loss. Values smaller than $s = 0.4$ further decrease the positive effects of the variational model, as the achievable variety of generated text is reduced. This phenomenon is referred to as posterior collapse (Lucas et al., 2019).

Combining the findings from evaluating the loss and latent space properties in Figure 3, we select $s = 0.4$ as scaling factor for further evaluating the VAT. This value corresponds to good reconstruction ability and isotropic representations which impacts both the generation of text as well as the performance on classification tasks. Also, the learning curves corresponding to $s = 0.4$ exhibit no overfitting any more (see Figure 2b).

---

[2]Different from Ethayarajh (2019), Figure 3a depict the plain intra-sentence and self similarity, i.e. not adjusted for isotropy.

---

**In this class, we will introduce several fundamental concepts.**

---

In this class, we will introduce several fundamental concepts.

---

*In this class, we will introduce several fundamental principles.*
*In this area, we will introduce several fundamental principles.*
*In this process, we will introduce several educational concepts.*
*In this class, I will introduce several fundamental principles.*
*In this progress, we will introduce private fundamental concepts.*
*In this class, we have received several fundamental concepts.*
*In this class, we will introduce several public projects.*
*In April 2013, we will introduce several fundamental issues.*
*In this class, we will find several fundamental principles.*
*In one class, she will introduce several fundamental concepts.*

---

Table 1: Variational Sampling. Example sentence (in **bold**) was used as input to the VAT, with the decoded mean representation and samples from the latent distribution (in *italics*).

---

**Generative models have shown great promise in modeling complex distributions.**

---

*advantages, results, achievements, quality, answers participation, moments, experience, passion, joy, benefits, interests*

---

Table 2: Variational Sampling. Example sentence (in **bold**) that was used as input to the VAT, with only the underlined token being sampled. Samples from the latent distribution in *italics*.

---

**I want to talk to you.**
I want to report.
I said:
She didn't want to say.
**She didn't want to be with him.**

---

**He was silent for a long moment.**
He was silent for a long moment.
He was my father.
It was my face.
**It was my turn.**

---

Table 3: Interpolation. First and last sentence (in **bold**) were given as input to the VAT. Intermediate sentences were obtained by token-wise linear interpolation between the two latent representations padded to the same length.

## 5 VARIATIONAL LANGUAGE MODEL

During training, the VAT encoder produces variants of the input tokens by drawing from the latent distribution. The VAT decoder has to be able to reproduce the original sequence given variational input. At test time, both approaches are possible: Decoding with the standard deviation set to 0, i.e. $z_t = \mu_t$, leads to a deterministic representation and is used for classification tasks. For sequence generation, decoding $z_t = \sigma_t \cdot \epsilon' + \mu_t$ allows to generate variants of the original input sequence, a process we refer to as variational sampling. Variational language modeling thus denotes the various ways of manipulations and computations in the latent space that are possible with the VAT's latent distributions. As examples, we discuss anecdotal results from variational sampling and interpolation.

Tables 1 and 2 illustrate the generating capabilities of the VAT with variational sampling. In the second line in Table 1, the VAT is able to reconstruct the original sequence (first line), which refers to decoding with $z_t = \mu_t$, i.e. without sampling. The following lines show exemplary variants of the input sentence obtained by randomizing the latent representations, displaying 12 variational samples. It is interesting that, while mostly maintaining the original sentence structure and original context, variants are found for almost all tokens. This gives an idea of the structure of the latent space and the contextuality of the representations. The approach is comparable to paraphrase generation, which is often obtained through back-translation (Sennrich et al., 2016), (Wieting & Gimpel, 2018) or by directly trained on supervised data in the form of paraphrase sentence pairs (Gupta et al., 2018).

Table 2 illustrates variational sampling for a single token, where the input sequence (in bold) is reconstructed according to $z_t = \mu_t$ except for the underlined token for which sampling is allowed. This setting is similar to a nearest neighbor search over a fixed corpus. The second line lists the obtained samples. By masking the underlined token, this approach is similar to gap filling (Donahue et al., 2020), (Wu et al., 2019). Variational sampling, both for tokens and sequences, can be useful for generating paraphrases in tasks such as dialog generation, question answering etc.

We also experimented with latent space interpolation similar to (Bowman et al., 2016), (Liu & Liu, 2019), (Li et al., 2020b). We linearly interpolate the latent representations of two sentences (padded

| | STS12 | STS13 | STS14 | STS15 | STS16 | STSb | SICK-R | Average |
|---|---|---|---|---|---|---|---|---|
| *Published in (Reimers & Gurevych, 2019)* | | | | | | | | |
| BERT-*start* | 20.16 | 30.01 | 20.09 | 36.88 | 38.08 | 16.50 | 42.63 | 29.19 |
| BERT-*mean* | 38.78 | 57.98 | 57.98 | 63.15 | 61.06 | 46.35 | 58.40 | 54.81 |
| *Our results* | | | | | | | | |
| MiniBERT-*start* | -2.19 | 0.87 | 1.27 | 0.81 | -3.46 | -2.41 | 0.61 | -0.64 |
| MiniBERT-*mean* | -1.14 | -3.83 | 1.63 | 0.21 | 7.25 | -5.3 | -1.73 | -0.42 |
| MiniBERT-*sum* | -0.50 | -1.54 | 0.60 | -0.72 | 0.11 | 1.21 | 1.67 | 0.12 |
| VAT-*start* | 17.45 | 12.10 | 14.74 | 20.09 | 31.14 | 30.70 | 37.15 | 23.34 |
| VAT-*mean* | 45.60 | 45.43 | 52.07 | 55.45 | 58.13 | 50.83 | 47.08 | 50.66 |
| VAT-*sum* | 45.60 | 45.43 | 52.07 | 55.45 | 58.13 | 51.56 | 48.88 | 51.02 |

Table 4: Spearman rank correlation between the cosine similarity of sentence representations and gold label similarity for different semantic textual similarity (STS) tasks without fine-tuning the underlying models to the target data.

to the same length) with three intermediate steps. Decoding the intermediate representations results in the sentences illustrated in Table 3 for two examples. Observing a smooth interpolation trajectory as in the examples is not always the case. It is possible that after decoding, the intermediate steps are mapped to the same sentence as seen for one sentence in the second example. This is especially true with increasing size of training data: The more training data, the more contextualized representations for the same word type that are mapped close to each other in latent space will exist.

## 6 SENTENCE REPRESENTATIONS

For the construction of sentence representations, we compare two pooling operations (average, sum). Additionally, the start token ("CLS") representation can be used as embedding for the whole sentence, but this approach has been shown to be inferior to mean pooling in semantic similarity tasks for BERT (Reimers & Gurevych, 2019). We denote the different approaches as VAT-*mean*, VAT-*sum* and VAT-*start*, respectively. We compare our model to the original BERT model (base) and, for a fair comparison in terms of model size, to a smaller variant with similar network dimensions than the VAT. BERT has $L = 12$ encoder layers and the model and representation dimension is $d = 768$. The smaller BERT model[3] (Turc et al., 2019), which we refer to as MiniBERT, has $d = 128$ and $L = 4$, the same size as the VAT. Both BERT and MiniBERT were trained on a much larger collection of datasets (Wikipedia, BookCorpus, CommonCrawl) than the VAT.

The sentence representations are tested using the SentEval (Conneau & Kiela, 2018) toolkit[4]. The toolkit evaluates static sentence representations on two different classes of tasks: semantic similarity and sentence classification tasks. Table 4 lists the correlation values of the different sentence representation methods on the semantic textual similarity (STS) tasks. For each pair of semantically similar sentences, the spearman correlation rank between the cosine of their latent representations and a human-labeled gold standard (between 0 and 5) is computed (no fine-tuning or transfer learning). The correlation of VAT-based sentence representations is on par with BERT-based representations which is surprising given the smaller model architecture and capacity of the VAT. For both VAT and BERT, start token based representations show less correlation than those obtained by average or sum pooling. Representations obtained from the MiniBERT model do not show any correlation at all, indicating that the BERT architecture requires a larger model size.

The classification tasks included in the SentEval toolkit comprise binary (MR, CR, SST2) and fine-grained (SST5) sentiment or polarity classification tasks, paraphrase detection (MRPC), natural language inference (SICK-E) and question-type classification (TREC) tasks. As paraphrase detection and natural language inference require the comparison of two sentences, the pair of input sentences is concatenated and separated by a special separation token to form a single input vector. For the classification tasks, we compare the performance of the VAT to that of MiniBERT in two ways: in a feature-based approach and in a fine-tuning approach.

---

[3]https://github.com/google-research/bert
[4]https://github.com/facebookresearch/SentEval

|  | MR | CR | SST2 | SST5 | TREC | MRPC | SICK-E |
|---|---|---|---|---|---|---|---|
| *Feature-based Approach* | | | | | | | |
| MiniBERT-*start* | 50.8/50.1 | 63.9/63.8 | 51.2/49.9 | 27.7/24.6 | 23.7/18.2 | 67.8/66.5 | 56.4/56.7 |
| MiniBERT-*mean* | 50.9/50.1 | 61.6/60.9 | 53.0/50.3 | 24.4/25.1 | 22.9/18.2 | 65.5/65.9 | 55.6/55.9 |
| MiniBERT-*sum* | 51.9/49.6 | 62.2/59.7 | 51.4/51.5 | 25.8/23.0 | 23.0/20.8 | 64.8/57.6 | 48.4/49.7 |
| VAT-*start* | 60.5/59.6 | 63.8/63.8 | **61.6/63.3** | 29.4/29.3 | 35.1/40.4 | 67.5/66.5 | 56.4/56.7 |
| VAT-*mean* | 61.0/60.2 | 63.9/63.8 | 61.4/60.7 | 30.4/31.0 | 44.1/48.6 | 67.5/66.6 | 59.2/58.5 |
| VAT-*sum* | **61.2/60.5** | **66.8/65.7** | 61.0/62.6 | **31.5/32.2** | **46.8/54.8** | **69.6/68.5** | **67.8/66.6** |
| *Fine-tuning Approach* | | | | | | | |
| MiniBERT-*start* | 72.8/72.1 | 64.4/61.9 | 80.6/83.2 | 36.7/36.1 | 32.1/38.8 | 68.9/66.5 | 56.4/56.7 |
| MiniBERT-*mean* | **76.0/75.6** | 73.4/65.6 | **81.0/83.3** | 38.0/37.9 | 72.4/73.2 | 71.8/68.6 | 58.2/58.1 |
| MiniBERT-*sum* | 73.8/70.7 | 75.4/67.6 | 70.0/81.6 | 35.8/32.8 | 70.8/69.6 | 68.9/63.0 | 57.6/44.3 |
| VAT-*start* | 74.5/74.6 | 76.3/72.5 | 82.7/82.6 | 37.8/38.6 | 82.7/87.2 | **70.0/69.9** | 61.6/61.3 |
| VAT-*mean* | 73.8/74.1 | 75.0/72.5 | 81.7/82.4 | **39.9/40.0** | 84.1/85.2 | 68.9/63.7 | **64.2/61.8** |
| VAT-*sum* | 72.9/73.2 | **78.6/74.5** | **82.6/83.3** | 40.7/39.9 | **84.7/88.2** | 69.7/68.3 | 62.0/60.8 |

Table 5: Dev/test accuracies on the SentEval transfer classification tasks. A single dense layer with nonlinear activation and Adam optimizer is used as classification layer. **Bold** values indicate the best test result for each task in the feature-based and fine-tuning approach.

Both presented approaches are different from how state-of-the-art models are trained for transfer tasks. Given the smaller model capacity, we do not expect on-par performance, but rather want to better understand the kind of information stored in the representations and the potential that comes with additional classification layers on top of the model. The comparison to MiniBERT serves as baseline.

For the feature-based approach, the sentence representations are extracted from the pretrained models and then passed to a single classification layer trained with an Adam optimizer (Kingma & Ba, 2015). Given the results in Table 5 (top), the sum of the individual token representations is best suited as standalone feature for sentence classification. As VAT-*sum* outperforms VAT-*mean* on all tasks, the dimensions of the latent representations seem to be well disentangled. With the exception of the SST2 task, the start representation yields lowest accuracy values suggesting that the first representations does not capture the full context of a sentence. MiniBERT, which is designed for fine-tuning rather than feature extraction, mostly does not reach the performance of VAT variants. The difference between MiniBERT and VAT is especially large for the TREC task.

For assessing the performance with a fine-tuning approach, we extend the underlying models by a single non-linear dense classification layer and optimize with Adam. For the VAT, we only reuse the encoder part to access the latent representations $Z$. For each task, we trained for five epochs Note that the MiniBERT results can differ from the results in the original publication (Turc et al., 2019), as we do not further tune the model or the training process.

In Table 5 (bottom), we see a performance leap for all of the models compared to their feature-based results. While VAT-*sum* was performing best for the feature-based approach, the method to produce sentence-level representations is not that decisive for the fine-tuning approach. The differences between *sum*, *mean* and *start* are not as great as for the semantic similarity tasks any more, neither for MiniBERT nor for VAT. It is interesting that MiniBERT is in lead for sentiment classification tasks, whereas VAT shows outstanding performance on the TREC (topic classification) task.

## 7 RELATED WORK

The architecture of variational autoencoders has been optimized for various natural language processing tasks by implementing various kinds of networks for encoder and decoder. The variational loss is applied to a sentence-level latent representation in all these models. Some of the networks do not contain a decoder any more, such that the latent representations are not interpretable.

Several models were suggested to solve common document or text classification tasks: Gururangan et al. (2019) use MLPs as encoder and decoder to learn latent vectors from bag-of-words inputs that are used for document classification. Mahabadi et al. (2021) compress BERT-based sentence

representations through a Gaussian prior distribution for transfer classification tasks. Deudon (2018) implements a siamese network architecture based on sentence representations from a VAE with Bi-LSTMs for semantic similarity classification.

Zhao et al. (2018) use a LSTM-based VAE for dialog generation. Miao et al. (2016) rely on MLPs in a VAE for both document modeling and answer selection. Wang & Wan (2019) introduce a Transformer-based VAE for story completion, where the missing plot is conditioned on the latent representation $z$ capturing the distribution of the story plot. Shu et al. (2020) propose a Transformer-based non-autoregressive model specifically for machine translation that incorporates a predictor network for the length of the target sentence.

Various network architectures have been proposed for the more general task of language modeling. Bowman et al. (2016) apply the variational loss on the last hidden state of a LSTM sequence-to-sequence model. In iVAE (Fang et al., 2019), an MLP produces sample-based distributions from an LSTM's hidden representation concatenated to random noise. Yang et al. (2017) apply an LSTM as encoder and a CNN as decoder in a language model. Liu & Liu (2019) add a feed-forward layer to a Transformer model to map the encoder outputs to a mean and variance vector which are then upsampled and passed to the decoder followed by an LSTM layer. OPTIMUS (Li et al., 2020b) extends this idea to large-scale language models. The encoder weights are initialized with pre-trained BERT weights, the decoder with those from a pre-trained GPT-2 model. The latent representation of the start token is used as sentence-level representation.

In contrast to these networks, the VAT's variational loss is applied at token-level which allows for direct manipulations in the latent space. Token-level regularisation has been previously applied to RNNs for language modeling (Li et al., 2020c). While language modeling is the training objective of the VAT, we also investigate the performance of the trained model on text classification tasks.

## 8 DISCUSSION AND CONCLUSION

The goal of the proposed VAT model is to obtain isotropic representations mapped to a smooth and complete latent space. The hyperparameter and scaling factor $s$ is directly optimized to meet these criteria. The effect can be seen especially for text generation, where the VAT is able to produce coherent sequences through manipulations directly in latent space. However, optimizing $\mathcal{L}_{REG}$ comes at the cost of $\mathcal{L}_{REC}$. Perfect reconstruction of the original input is not possible at all times any more.

Reducing the representations to a single sentence-level vector preserves contextual information. Semantic similarity tasks demonstrate that the smaller model capacity of the VAT is sufficient to capture semantic information that is on a par with larger BERT architectures. However, models explicitly trained to improve the performance on semantic similarity tasks are out of reach.

The VAT is able to produce more robust standalone sentence-level features when compared with MiniBERT for the feature-based classification approach. VAT outperforms the similar-sized MiniB-ERT also when being fine-tuned on sentence classification tasks, but is far from the state-of-the-art performance known from BERT-sized models with more sophisticated training procedures. Interestingly, VAT shows its peak performance for the topic classification task TREC. In our setup, the VAT captures topic-related information even better than sentiment.

Tasks involving the comparison of two sentences (MRPC, SICK-E), represented by a single concatenated vector, cannot be successfully solved by either VAT or MiniBERT. Some model variants (MiniBERT included) only learn to predict the most frequent class. The reason could be the smaller model sizes that are incapable of capturing the information of two concatenated sentences, as especially the VAT was never trained on such a setting. Instead of a concatenated vector, the comparison task could be solved using a siamese network architecture, similar to Deudon (2018).Optimizing the regularization loss at token-level could even allow the training of a multilingual model or a translation model, given sufficient training data.

REPRODUCIBILITY STATEMENT

All data used for training and evaluation of the model are publicly available. The WMT19 dataset was obtained through the tensorflow library at `https://www.tensorflow.org/datasets/catalog/wmt19_translate`, all other datasets are available through the SentEval toolkit at `https://github.com/facebookresearch/SentEval`. The source code for instantiating and training our model as well as for decoding and generating variational samples is made available as supplementary material. Hyperparameters and training settings are described in Section 4 and – in more detail – in Appendix A.

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

## A  APPENDIX

### MODEL DETAILS

The implemented VAT consists of $N = 4$ encoder and decoder layers, respectively, with $H = 8$ attention heads each. The encoder and decoder architecture including the positional embedding is the same as in the original Transformer (Vaswani et al., 2017) and can be seen in Figure 1. The model dimensionality $d = 128$, the dimension in the feed forward layers is $512$. Dropout during training is set to $0.1$. The Noam optimizer Vaswani et al. (2017) operates with $50,000$ warmup steps. A logistic annealing function Bowman et al. (2016) with $50,000$ warmup steps and an initial value of $0.00025$ is applied to weight the regularization term of the loss function.

The VAT is trained on the train split of the WMT19 de-en dataset Barrault et al. (2019) using English sentences only. The data is tokenized using subword tokenization with a target vocabulary of $2^{15}$ tokens. Batch size is set to $128$. Decoded sentences that are presented as results are obtained through beam search with beam size $5$.

ENERGY CONSUMPTION

Experiments were conducted using a private infrastructure, which has an estimated carbon efficiency of 0.197 kgCO$_2$eq/kWh. A single epoch takes 20 hours of computation on a RTX 2080 Ti (TDP of 250W) GPU. For the training of a VAT model for 3 epochs, total emissions are estimated to be 0.99 kgCO$_2$eq according to the ML impact calculator Lacoste et al. (2019) available at:

`https://mlco2.github.io/impact#compute.`

