# OpenReview forum: "Isotropic Contextual Representations through Variational Regularization"
_ICLR.cc/2022/Conference — ICLR 2022 Submitted_

### Official Review · Reviewer_j72L · 2021-10-29

**Correctness:** 3
**Technical Novelty And Significance:** 4
**Empirical Novelty And Significance:** 3
**Recommendation:** 6
**Confidence:** 3

**Main Review:**

**Strengths**

- Clear description of the proposed model.
- Extensive experiments show the usefulness of various design choices (e.g., scaling factor) of the proposed method.

**Weaknesses**

- For the evaluation of isotropy, currently there seems to be only those of VAT (with different hyperparameters). Some comparisons to other models (esp: MiniBERT_*) would better support the effectiveness of VAT in isotropy. This way the readers would have an idea of how the improvements of VAT relates to the improvements in classification tasks.
- The evaluation on interpretability can be deeper. Section 6 has some examples, but additional elaboration would be appreciated.
- There are already a lot of evaluations, but various virtues of the models can be connected by further evaluations. E.g., how do the isotropy and the performance on the transfer classification tasks relate?

**Comments and questions**

- Page 1 in the abstract "prior distribution". What does the prior refer to?
- Page 2, equation 2: is the norm L1? (Or L2?)
- Section 5.1: Did you normalize the self-similarity and intra-sentence similarity like the description of Ethayarajh (2019)?
- Also in section 5.1: Is the anisotropy the average cosine similarity?
- Section 6 briefly mentions the interpretability of the language model by presenting some samples from the latent distributions. In what way are the examples special?
- Adam optimizer: Please cite the paper proposing this algorithm.
- Several other papers use variational methods in language modeling. Worth mentioning in the related work section:

Yang, Zichao, et al. "Improved variational autoencoders for text modeling using dilated convolutions." *International conference on machine learning*. PMLR, 2017.

Li, Ruizhe, et al. "Improving variational autoencoder for text modelling with timestep-wise regularisation." *COLING*, 2020.

Fang, Le, et al. “Implicit Deep Latent Variable Models for Text Generation.” *Proceedings of the 2019 Conference on Empirical Methods in Natural Language Processing and the 9th International Joint Conference on Natural Language Processing (EMNLP-IJCNLP)*, Association for Computational Linguistics, 2019, pp. 3946–56.

**Summary Of The Paper:**

Provide a brief summary of the paper and its contributions.

This paper proposes Variational Auto-Transformer to encourage isotropy in the latent representation space. The resulting encoder-decoder architecture allows interpretable embeddings. On various tasks, VAT shows better performances than other language models.

Contributions:

- A novel architecture based on Transformer with a token-level variational loss.
- Extensive evaluations on the effectiveness of Transformer.

**Summary Of The Review:**

Proposes a novel method and clearly elaborates it. Extensive experiments verifies the effectiveness, as well as the design choices.

---

> ### Author Response · Authors · 2021-11-18
> **Response to Reviewer j72L**
>
> Thank you for your helpful comments!
>
> Concerning the similarity measures based on Ethayarajh (2019), we now clarify that we compute the average similarity for all, isotropy, self-similarity and intra-sentence similarity, and updated the corresponding paragraph (see subsection 4.2). Different from Ethayarajh (2019), we do not adjust for isotropy (normalization), as the measures are more directly comparable without adjustment. This is now also stated in a footnote (on page 5).
>
> We included references [1], [2] and [3] in the related work section (section 7).
>
> Evaluating the relationship between isotropy and the performance on transfer classification tasks would be an interesting investigation, but is currently left to future work.

---

> > ### Comment · Reviewer_j72L · 2021-12-01
> > **Thank you for the reply**
> >
> > Thank you for the reply, updating, and clarification.

---

### Official Review · Reviewer_Fv5U · 2021-10-31

**Correctness:** 2
**Technical Novelty And Significance:** 3
**Empirical Novelty And Significance:** 2
**Recommendation:** 3
**Confidence:** 4

**Main Review:**

The idea of using a variational autoencoder to address the degeneracy of the representations of BERT-like models is sensible, intuitive, and appears to be novel. However, the paper contains significant weaknesses:

- In the experiments, it is not clear that MiniBERT is the fairest baseline.
    - Although the encoder of VAT may have the same number of parameters as the MiniBERT model, VAT also has a decoder during training. Therefore it can be argued that a more appropriate baseline would be a BERT model which has the same total number of parameters as the entire VAT model (decoder included).
- Overall, the writing lacks clarity.
    - Section 2, paragraph 3
        - It is stated that the "highly discrete latent space lacks completeness and is of limited use for advanced NLP applications"? Are the authors able to provide justification for this? One could argue that BERT (+ variants) are "over-complete" autoencoders as described, and they have revolutionized NLP.
    - Section 2.1, paragraph 3
        - At the end of this paragraph, should it not be the Cholesky root of $\Sigma$ which multiplies $\epsilon$?
    - Equations (2) and (4)
        - The L2 loss notation (with the double vertical lines) can be misleading. What if $x$ is discrete? It would be clearer to say something along the lines of $x$ is distributed according to some distribution $p_{X}(\cdot)$ and then $L_{REC} = -\log p(dec(enc(x)))$ or $L_{REC} = -\log p(dec(z))$.
    - Section 3
        - The paper would flow better with Section 3 appearing after Section 4 instead of before. As it currently stands, it is difficult for the reader to infer the differences between your work and the various prior methods referred to. This would be easier if your method had already been described.
    - Section 4
        - The authors don't ever explicitly define how the model is constructed / what the objective actually is. Based on Equation (4) and the ensuing description in Section 4, the reader is left to assume that the objective must be $\sum_{t} \mathbb{E}_{q(z_{1:T}|x_{1:T})}[\log p(x_{t}|x_{1:t-1},z_{1:T})] - D_{KL}[q(z_{t}|x_{1:T}) || p(z_{t})]$ but this is never actually stated, nor is a graphical model provided.
    - Section 4.1, paragraph 1
        - The KL annealing suggested by Bowman et al. (2016) isn't intended to combat overfitting, but rather the so-called 'KL collapse' phenomenon. This is where the latent variable gets ignored because the model $p(x|z)$ is able to reconstruct $x$ while ignoring $z$, and so is able to set $q(z|x) = p(z)$ for all $x$ such that $KL[q(z|x)||p(x)] \rightarrow 0$. The curves in Figure 2 only seem to start at iteration 50,000 but as shown, they don't indicate that the model is suffering from KL collapse. If this is this what happens in the experiments, the explanation for using KL annealing should be amended.
    - Section 5.1, paragraph 5
        - As mentioned above, posterior collapse is when the regularization loss is low and reconstruction loss is low, not when regularization loss is low and reconstruction loss is high. It is not clear that the authors have understood this phenomenon.

**Summary Of The Paper:**

This paper presents a variational autoencoder model for learning representations of tokens in a sequence of text. The authors build upon prior work which argues that the point-estimate representations learned by Transformer-based language models are degenerate, i.e. they only populate a narrow subspace. The authors address this problem by learning Gaussian-distributed representations (instead of point-estimates) which are regularized using a KL divergence loss. Empirical results on classification tasks are competitive with point-estimate baselines.

**Summary Of The Review:**

The core idea presented in this paper is strong, but the lack of an appropriate baseline and the significant rewriting required make it unsuitable for acceptance.

---

> ### Author Response · Authors · 2021-11-18
> **Response to Reviewer Fv5U**
>
> Thank you for your valuable comments! We appreciate that the reviewer sees our model as intuitive and, in addition to the general remarks, we would like to add:
>
> **RE: In the experiments, it is not clear that MiniBERT is the fairest baseline.**
>
> When VAT is used for classification tasks, only the encoder part (that generates the latent representations z) is reused. Thus, we chose to compare to MiniBERT with the same number of encoder layers and the same model dimension.
>
> Concerning the remaining remarks, we hope to have successfully updated formulas and explanations and want to explicitly point the reviewer to
>
> - formulas (5) and (6) that now contain details on how to compute the objective for the full model
> - Subsection 4.1 (second-to-last paragraph on page 5) where the posterior collapse is now attributed to the appropriate loss characteristics in Figure 3b and a more appropriate first paragraph in Subsection 3.1

---

> > ### Comment · Reviewer_Fv5U · 2021-11-22
> > **Response to authors**
> >
> > Thanks for the response.
> >
> > Regarding the choice of baseline: the vast majority of the computation in such models is in pretraining the upstream model (autoencoder in your case), while training the classifier is relatively cheap. Therefore it is still not clear that MiniBERT is the fair baseline in this setting.
> >
> > In Equation (5), what is the reason for using the squared loss (equivalent to log likelihood under a Gaussian distribution, using the embeddings as the observations) rather than the more typical log likelihood under a Categorical distribution, using the tokens as the observations [1, 2]?
> >
> > [1] Samuel R. Bowman, Luke Vilnis, Oriol Vinyals, Andrew Dai, Rafal Jozefowicz, Samy Bengio. Generating Sentences from a Continuous Space. CoNLL, 2016.
> > [2] Jacob Devlin, Ming-Wei Chang, Kenton Lee, Kristina Toutanova. BERT: Pre-training of Deep Bidirectional Transformers for Language Understanding. NAACL, 2019.

---

> > > ### Author Response · Authors · 2021-11-24
> > > **Response on model choices**
> > >
> > > Thank you again for your comments and deepening questions!
> > >
> > > We can follow your argumentation concerning the baseline model size. Ours is from the perspective of the classification task: Given two pretrained models that produce representations suitable for a final classification layer, we have (approximately) the same number of parameters to fine tune.
> > >
> > > Concerning equation (5): Given the static positional encoding that is applied for the VAT in the same way as for the original Transformer, we consider the input observations to be continuous. This is different from BERT-models, where also the positional embedding is learned and the only observable input are the tokens.

---

### Official Review · Reviewer_ZoL1 · 2021-11-02

**Correctness:** 2
**Technical Novelty And Significance:** 2
**Empirical Novelty And Significance:** 2
**Recommendation:** 5
**Confidence:** 4

**Details Of Ethics Concerns:**

No concerns.

**Main Review:**

The strength of the paper:
1. The discussion on selecting proper $sigma$ and how it affects the training, provides a good hint on selection of the hyper-parameters.
2. The variational sampling and interpolation is interesting.

The weakness of the paper:
1. Some part of the paper is not clearly written, e.g. Table 3, for miniBERT, I guess they use a relative number ?
2. The major flaw is the experiments. Although the paper mentioned it is not going to provide on-par performance with the SOTA BERT, it is promissing that since the small-sized model can achieve better performance, potentially also applies to larger models. This is a handwaving claim and cannot justify the much worse performance (Table 4) compared with those smaller models such as DistilBERT or TinyBERT [2] (also 4 layers). This makes it very hard to judge the contribution of this paper, as it does not obtain even a close-to-SOTA (e.g. for MRPC, 70 in this paper vs 85 in tinyBERT) performance. If comparable results could be obtained, e.g. both on 12-layer fully trained transformers, then I can believe the variational autoencoder layer would no hurt the performance.

Minors:
1. Bottom of page 2, before equation (4), where the reparameterized sample $z$ does not equal to $\Sigma \epsilon + \mu$, but rather $L^T \epsilon + \mu$, where $L^T L = \Sigma$.
2. The isotropy talked in this paper is referring to the latent code $z$, rather than the hidden representations $h$ which is most common in literatures (e.g.  Ethayarajh, 2019)
3. The idea is very similar to the variational information bottleneck, which is also used in language models e.g. [1], as a regularizer in model training/fine-tuning.

[1] Mahabadi, R. K., Belinkov, Y., & Henderson, J. (2021). Variational Information Bottleneck for Effective Low-Resource Fine-Tuning. arXiv preprint arXiv:2106.05469.

[2] Jiao, Xiaoqi, et al. "TinyBERT: Distilling BERT for Natural Language Understanding." Proceedings of the 2020 Conference on Empirical Methods in Natural Language Processing: Findings. 2020.

**Summary Of The Paper:**

This paper proposes to add a variational loss for each token, in the transformer architecture, to increase isotropy of deep language models. To achieve stable training, the paper proposed to adjust $\sigma$ of the prior Gaussian distribution, and obtain a smoother training curve. The paper evaluated the proposed method from different aspects, including variational sampling, interpolation, semantic textual similarity and semantic classification tasks.

**Summary Of The Review:**

The idea is similar to variational information bottleneck. The main concern is the experiments, especially table 4, that is too far behind other state-of-the-art models that have a similar size.

---

> ### Author Response · Authors · 2021-11-18
> **Response to Reviewer ZoL1**
>
> Thank you for your valuable comments! We appreciate that the reviewer liked the variational sampling and interpolation capacities of our model and like to address the remarks:
>
> **RE: Some part of the paper is not clearly written, e.g. Table 3, for miniBERT, I guess they use a relative number ?**
>
> We hope that the respective parts are clearer in the revised version but are not sure what the reviewer mean by "use a relative number". Results in Table 3 are obtained in the same way for all methods and the Spearman correlation for MiniBERT is as reported.
>
> **RE: The isotropy talked in this paper is referring to the latent code z, rather than the hidden representations h which is most common in literatures (e.g. Ethayarajh, 2019)**
>
> We use the latent code z for classification tasks (the decoder is not needed any more), thus we argue that in the case of VAT, the latent code is also the hidden representation.
>
> **RE: The idea is very similar to the variational information bottleneck [1].**
>
> We have included VIBERT [1] in the related work section. However, VIBERT is targeted at classification tasks and is not able to decode the representations any more. Tasks such as paraphrase generation, which is possible with our VAT, cannot be solved using VIBERT.
>
> Lastly, we'd like to point out that MiniBERT as used in our paper is a pretrained small BERT model without distillation, which is why the results on benchmark tasks is different from tinyBERT [2]. With VAT, it is not our intention to improve the state-of-the-art on text classification tasks, but rather to evaluate how well the variational model, that allows for sentence generation and paraphrasing, performs on standard classification settings.

---

> > ### Comment · Reviewer_ZoL1 · 2021-11-19
> > **Thanks for your response**
> >
> > Dear authors,
> >
> > Thank you for your response. So the Table 3, miniBERT, the correlation is around 0, while other models yield a much higher correlation. I thought the miniBERT results are relative (a dellta value, compared to the original BERT model). Thanks for clarification.
> >
> > However, the concern on the worse performance is not very well addressed. Given the time limit, I don't think performing an additional experiment on applying the variational layer to a full BERT model, could be finished. I recommend adding a commparable model and demonstrate on both variational sample comparison (with some statistics rather than individual samples), and the classification task, the proposed method could yield close/better performance. That will make the paper much stronger.

---

> > > ### Author Response · Authors · 2021-11-23
> > > **Response on model comparison**
> > >
> > > Thank you again for your response. Our main aim is to generate isotropic representations (thus applying the variational loss on token-level) and demonstrate their suitability for sentence generation through directly manipulating the latent space. Adding a variational loss to MiniBERT (a similar-sized model to VAT) to achieve the same variational sampling ability is not feasible, as BERT-like models consist of an encoder only, i.e. decoding is not possible. Additionally, other variational models (mostly LSTM-based) in the literature are optimized for either language generation or classification. This is why we do not directly compare with other variational models in the experiments: For the generation part, the reader is referred to anecdotal results from other models in the literature (now better referenced throughout section 6). For the classification part, we compare to a "normal" model without variational loss in order to contextualize the results.
> > >
> > > While implementing models and experiments on a large scale would certainly be interesting, it is beyond scope for this paper as our focus is on smaller-scale settings. We have eliminated the somewhat misleading claim concerning larger models from the conclusion.

---

> > > > ### Comment · Reviewer_ZoL1 · 2021-11-29
> > > > **Response to authors**
> > > >
> > > > Thanks again for the clarification. However, I will stick to my original rating, as the two major concerns still remain unclear: 1. The benefit of increasing isotropy? There is no such comparison with other variational models that do not promote isotropy. 2. The miniBERT baseline is weak, which is also pointed by other reviewers. Addressing these two issues will strengthen this paper by a large margin.

---

### Official Review · Reviewer_pcez · 2021-11-02

**Correctness:** 2
**Technical Novelty And Significance:** 2
**Empirical Novelty And Significance:** 1
**Recommendation:** 3
**Confidence:** 4

**Main Review:**

**Strengths**

Isotropy of word embedding space has been a topic of strong interest to the NLP community in recent years, and it is reasonable to adopt VAE as the underlying method.

**Weaknesses**

Unfortunately, almost all of the experiments have some flaws that are difficult to ignore, and the manuscript might not show the superiority of the proposed method.

1. Regarding variational sampling: Various methods have been proposed that sample sentences with similar meanings to input sentences, such as back-translation [1,2], text infilling [3], and VAE [4]. Since there is no comparison between these existing methods and the proposed method in the manuscript, we must say that the superiority of the proposed method is unknown. Additionally, only one example is given for the output of the proposed method, and we cannot shake the concern that this is champion data. The persuasiveness of the paper will be improved if the proposed method is appropriately compared to existing methods through descriptions and experiments.
2. Regarding interpolation: There is no mathematical or procedural description of the computation and no citations. It must be said that the manuscript lacks self-containedness and reproducibility. The reader would benefit from a clear description of the algorithm. In addition, as the authors stated, if it does not work well in general, then it is somewhat inadequate to convey the merits of the proposed method.
3. Regarding sentence representation: There are two problems with the comparison with MiniBERT.
    - First, as mentioned by the authors, the training corpus is different for the proposed and baseline models. Two variables are simultaneously changed, the domain of the data and the model, and this cannot be said a control experiment. Furthermore, the community has developed the learning tool for BERT [5], and comparative experiments can be performed easily.
    - Second, the NLP community rarely uses BERT-mini in pure research or practical applications. Therefore, even if the proposed method outperforms BERT-mini in controlled experiments, it is hardly validated that the proposed method is a valuable model. The empirical persuasiveness of this study would be significantly improved by increasing the parameters of the proposed model and comparing it with commonly used models such as BERT-base and BERT-large.

---

- [1] Sennrich et al, Improving Neural Machine Translation Models with Monolingual Data (ACL 2016)
- [2] Wieting and Gimpel, ParaNMT-50M: Pushing the Limits of Paraphrastic Sentence Embeddings with Millions of Machine Translations (ACL 2018)
- [3] Donahue et al, Enabling Language Models to Fill in the Blanks (ACL 2020)
- [4] Gupta et al, A Deep Generative Framework for Paraphrase Generation (AAAI 2018)
- [5] Wolf et al, Transformers: State-of-the-Art Natural Language Processing (EMNLP 2020, System Demonstrations)


**Summary Of The Paper:**

To obtain an isotropic word embedding space, the authors proposed a new transformer-based autoencoder in which each word is represented as a normal distribution, and a variational loss is used to make it closer to an isotropic normal distribution. By carefully adjusting the weights of the variational loss (magnitude of variance), the learning converged, and an isotropic latent representation was obtained. They also tried to show some of the advantages of the proposed approach:
1. The proposed model can variationally sample sentences with slightly different meanings from the input sentences.
2. In some cases, the proposed model can generate a semantic "interpolation" between two given sentences.
3. Solving sentence-level tasks using word representations obtained by the proposed model outperforms MiniBERT.

**Summary Of The Review:**

I have no objection to the importance of the theme or the validity of the idea. However, almost all of the experiments did not show the superiority of the proposed method, and it would be difficult to accept it to ICLR, a leading conference.

---

> ### Author Response · Authors · 2021-11-18
> **Response to Reviewer pcez**
>
> Thank you for your valuable comments! We hope to be able to clarify your concerns:
>
> In section 5, we have now explained the approach of variational sampling in more detail and how it is related to the other paraphrase generating methods that you mentioned. With variational sampling, paraphrasing both a whole sentence/sequence (as with back-translation) or individual positions within a sequence (as with gap filling) is possible. Different from VAE [4], no supervised dataset is needed. However, we think a comparison of the methods based on exemplary output is subjective. Comparison based on BLEU score would require a different training regime for our model.
>
> Interpolation is done by linearly interpolating the individual tokens of the two sequences (that were padded to have the same length), with a fixed number of three intermediate steps. Similar to variational sampling, interpolation would also work for a single (or only some) token as well. The approach is now described in more detail verbally (in Section 5). Due to the limited space, we couldn't add an algorithmic description/formula.
>
> It is not our intention to improve the state of the art on text classification tasks, but rather to demonstrate that the variational model, that allows for sentence generation and paraphrasing, is still applicable in standard classification settings. Thus, we refrain from training MiniBERT on the same dataset, as we want to compare our model to the practical setting where one would pick an available pretrained model.

---

> > ### Comment · Reviewer_pcez · 2021-11-29
> > **Response to authors**
> >
> > Thank you for your response!
> >
> > I believe that the overall description of the manuscript has been improved. Since connections to related research have been made clear, the manuscript is now easier for the reader to judge the contribution of the proposed method to the research field.
> >
> > However, unfortunately, the main concerns have not been addressed, so I would like to keep the rejection rating.
> >
> > 1. For variational sampling, the superiority of the proposed method cannot be judged without comparing it with other methods. Even if it is "subjective," a qualitative comparison would be possible.
> > 2. There are few descriptions of the interpolation, and it is almost impossible for the reader to reproduce the experiments.
> > 3. Regarding the comparison with MiniBERT, the following two points remain unsolved: (1) it is not a controlled experiment, and (2) the size of the model is too small for practical use.

---

### Author Response · Authors · 2021-11-18
**General Comments and Revision Summary**

Dear reviewers,

thank you very much for your efforts, your mindful comments and suggestions. Allow us some general remarks, before addressing your concerns individually.

- Section 3, the related work section, was moved to the end of the paper (before the discussion section). Unfortunately, this will prevent useful output from pdfdiff. We thus directly reference sections/paragraphs in our responses to identify changes. The references indicate the position in the new document.
- The abstract and the introduction is now better aligned to our aim of proposing a model that improves latent space characteristics such as isotropy and completeness.
- We have improved the readability and conciseness of the formulas throughout the manuscript. Section 3 now contains details on how to compute the loss function at token-level.
- We have updated Section 5 Variational Language Model substantially by adding more examples (Table 2 + 3) and more detailed explanations of the experimental setting.
- We removed the results of additional benchmark models in Table 4 (the STS task). It is not our intention to improve the state of the art on these tasks, but rather to compare the performance of our model with commonly used, pretrained models of comparable size. We hope the presentation of the results now better reflects this intention.
- The related work section (Section 7) is revised and also includes the references suggested by the reviewers.
- All these efforts have forced us to shorten other parts of the paper which lead to minor rephrasing at various places.
- We changed the xtick labels in Figure 2 as they were misleading.

---

### Decision · Program_Chairs · 2022-01-20

**Decision:**

Reject

**Comment:**

This paper develops a variational auto transformer model (VAT), a VAE based on the transformer (encoder-decoder) architecture designed to provide isotropic representations by adding a token-level loss for isotropy. All the reviewers agree that this is a novel architecture with a valid and interesting goal behind it.

Reviewers varied somewhat on their impressions of the paper, but none were strongly positive on accepting it. I think the strongest and most aligned concerns were from reviewers ZoL1 and pcez. They both feel that the experiments do not convincingly demonstrate what is required. It would be good to better establish the success of variational sampling and the usefulness of isotropic representations. I would think that even a page of examples in the appendix, contrasting sampling by various methods, would add a lot of information to what is presented here. It would be even better to have experiments showing the relation between improved isotropy and improved task performance (suggested by j72L). Both reviewers are concerned about the small model and weak results and whether these results would extend to larger models that people actually use. While on the one hand, controlled comparisons are valuable, it is also true that people in NLP routinely like to see results on models of a reasonably competitive size. In practice, for 2019-2021, it seems that people regard having models of BERT-base size as the "reasonable" small size that they will accept and for which there is reasonably good performance and lots of available empirical results. Transformers directly trained with very few layers do not perform that well. Reviewer pcez is also concerned about the change of the data set in the MiniBERT comparison, which seems valid, and reviewer 5v5U is concerned about what's fair in terms of parameter counts.

This paper needs further work with larger and more careful experimental comparisons to meet the needed level of experimental rigor to be convincing. The authors were not able to iterate sufficiently quickly to achieve this during the ICLR reviewing period, so it seems best that the paper be rejected for now, and the authors look to subsequently submit a more developed version of this work.